# Assessment of Dynamic Bayesian Models for Gas Turbine Diagnostics, Part 1: Prior Probability Analysis

**Valentina Zaccaria \*, Amare Desalegn Fentaye and Konstantinos Kyprianidis**

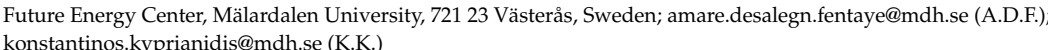

Future Energy Center, Mälardalen University, 721 23 Västerås, Sweden; amare.desalegn.fentaye@mdh.se (A.D.F.); konstantinos.kyprianidis@mdh.se (K.K.)
\* Correspondence: valentina.zaccaria@mdh.se

**Abstract:** The reliability and cost-effectiveness of energy conversion in gas turbine systems are strongly dependent on an accurate diagnosis of possible process and sensor anomalies. Because data collected from a gas turbine system for diagnosis are inherently uncertain due to measurement noise and errors, probabilistic methods offer a promising tool for this problem. In particular, dynamic Bayesian networks present numerous advantages. In this work, two Bayesian networks were developed for compressor fouling and turbine erosion diagnostics. Different prior probability distributions were compared to determine the benefits of a dynamic, first-order hierarchical Markov model over a static prior probability and one dependent only on time. The influence of data uncertainty and scatter was analyzed by testing the diagnostics models on simulated fleet data. It was shown that the condition-based hierarchical model resulted in the best accuracy, and the benefit was more significant for data with higher overlap between states (i.e., for compressor fouling). The improvement with the proposed dynamic Bayesian network was 8 percentage points (in classification accuracy) for compressor fouling and 5 points for turbine erosion compared with the static network.

**Keywords:** gas turbine diagnostics; dynamic Bayesian network; probabilistic diagnostics

## 1. Introduction

Reducing maintenance costs is one of the most pressing concerns for gas turbine owners. The goal of shifting to predictive maintenance to minimize unnecessary stops has driven the progress in diagnostics and prognostics in the last several decades. In particular, great advancements in computational power and machine learning techniques have led to a wide set of applications for gas turbine diagnostics [1–3]. Advantages of such data-driven methods include robustness against measurement noise, fewer sensor requirements, and no knowledge requirement of proprietary information [4]. However, they suffer mainly from data imbalance, i.e., a lack of a sufficient number and spread of fault samples in the historical data [5].

Data used in the diagnostic process are inherently uncertain due to inherit measurement errors (noise and bias). For this reason, probabilistic models such as Bayesian networks (BNs) [6] and probabilistic neural networks [7], or methods based on fuzzy inference [8], offer an effective tool for diagnostics. Their performance has often surpassed that of deterministic models [9–11]. One advantage of BNs compared to other data-driven methods is their ability to handle missing data and to integrate expert knowledge when historical data are inconsistent or incomplete [12]. To overcome the limitations related to model complexity, when multiple faults have to be analyzed simultaneously, hierarchical models have been proposed. Hierarchical Bayesian models were proven to successfully detect various sensor faults [13] and to isolate single and multiple faults also in the presence of sensor biases [14]. Despite their promising features, BNs have been employed mostly for fault isolation. The goal of tracking degradation evolution is commonly achieved with the help of physics-based models, Kalman filters, or particle filters [15–17]. These methods are considered more suitable for phenomena that vary over time, such as gradual deterioration.

A traditional Bayesian network represents contemporaneous dependencies, i.e., relationships between variables that exist at the same time. These can be extended, however, by adding the influence of time to construct a dynamic Bayesian network (DBN) [18]. In a DBN, also called a temporal BN, dependencies between present variables and past variables are made explicit. Therefore, these improved models are most suitable for those processes that are highly time-dependent or where a priori knowledge is affected by process dynamics [19]. Dynamic BNs have been successfully implemented for the fault diagnosis of various complex engineering systems [20–22]. The applications on gas turbine diagnostics are, on the contrary, still limited. In gas turbines, gas path components are subject to continuous deterioration due to gradual wear and tear, material deposition, and materials corrosion [23]. These degradation phenomena are better described with time-dependent models, since their severity is expected to worsen over time and one mechanism can often initiate or aggravate others. For example, it was suggested that compressor fouling could reduce the creep life of turbine blades, implying that previous knowledge on compressor conditions can be necessary for reliable predictions of current turbine status [24]. With this purpose, authors in [18] implemented a DBN for estimation of different degradation phenomena and fault propagation in gas turbines. A model of process degradation and fault dependencies was combined with current observations to infer possible degradation paths and perform risk assessment. In this way, a DBN allows us to incorporate more information useful for diagnostics than Kalman filters or particle filters. A better understanding of the advantages and disadvantages of the compared methods can be extracted from Table 1.

Ageing models are another application that can benefit from DBNs. In [25], the authors present the use of hierarchical BNs for failure rate estimation, which utilize a multi-stage prior probability as a function of certain hyperparameters. These hyperparameters may be, for example, geometrical characteristics or material properties that are associated with inherent uncertainty and may vary over time. Time-dependent failure rate (e.g., increasing over time) or the effect of maintenance actions can be considered in DBNs to estimate the correct prior probability distribution [25,26].

The prior probability distribution in BNs has an important effect on the results' accuracy, as the posterior probability is proportional to the product of prior probability and conditional probability. To date, there is no single best practice on how to define the prior probability of component conditions in a gas turbine. This is, in principle, built from experience, reasonably assuming that health parameters are likely to be at the design condition in a brand-new machine, and even for an aged machine, large deviations from the design condition are rare. However, when this information is solely extracted from operational data, it is possible that some rare fault events are overlooked because they have not occurred yet or sufficient data are not available. Therefore, one has to make other assumptions; for example, the authors in [14] made the conservative assumption of uniform prior distribution. By using a DBN, a more realistic prior probability function can be defined with time-dependent parameters. However, this has not been systematically assessed in the literature.

In addition, all the analyzed work from the open literature demonstrated the successful application of BN-based diagnostics on a single machine. However, due to the scarcity of fault samples in the history of a single machine, training data often have to be gathered from an entire fleet. Engine serial deviations and different operating or maintenance histories have an effect on the probability distributions selected for the BN model. Previous work on DBNs did not assess the benefits of DBNs over static BNs when fleet data are used for training and significant scatter is present. Furthermore, applications to gas turbine diagnostics are still too limited to draw a conclusion about the benefits of DBNs and what model is more suitable. In this paper, the use of a DBN for diagnostics of a power generation gas turbine is assessed, considering highly scattered fleet data to train the network. The work is divided into a two-part paper. In this Part 1, different approaches in setting up the prior probability distribution are compared in terms of accuracy of the compressor and

turbine degradation estimation. In Part 2, the BN models will be applied to synthetic data and real field data to diagnose multiple fault scenarios and discriminate between gradual, time-dependent degradation and abrupt faults [27].

**Table 1.** Summary of different methods used for fault diagnostics in gas turbines.

| Method | Typical Implementation | Benefits | Disadvantages |
|---|---|---|---|
| Physics-based models | Degradation tracking [1] Abrupt faults [15] | Full understanding of physical, thermal, and aerodynamic nature of the engine behavior. Qualitative and quantitative assessment of the health status of gas path component(s) is possible using measurement deviations. Multiple faults diagnosis. | Knowledge about component characteristic changes due to different faults required. The reliability is dependent on the fault magnitude. Large number of sensors required. Sensitive to measurement uncertainty. |
| Kalman filter (KF) | Degradation tracking [16] | Accurate estimations for linear problems. Low computational complexity. Measurement uncertainty is considered during diagnosis. The actual sensor noise can be represented by white Gaussian distribution. | Even the extended KF based methods can only handle problems with a limited amount of non-linearity. The effectiveness is affected by the unknown performance deterioration and measurement noise covariance matrices. Choice of appropriate covariance matrix is challenging task. Smearing effect can be present. |
| Particle filter | Degradation tracking [17] | Can be used to model multivariate, dynamic processes. More accurate than KF variants for non-linear systems. Coping with measurement uncertainty. | A large number of samples is required; hence, the computation can be heavy. Large number of sensors required. |
| Neural networks | Degradation tracking Abrupt faults [1] | Suitable for non-linear problems. Training can be done by means of information extracted from performance data without detailed knowledge of the gas path system. Multiple faults diagnosis. Measurement uncertainty can be considered. Suitable for problems with limited number of sensors. | Great amount of data needed for training representing the full operating envelope. Sensitive to class imbalance problems (when sufficient faulty data are not available) Retraining required after overhaul. Full understanding of physical and thermodynamics behavior is not possible (black-box model). |
| Fuzzy logic | Abrupt faults [8] | Model-free, knowledge of the process not required. Capable of generalizing from examples. Coping with measurement uncertainty. Suitable for non-linear problems and multiple faults diagnostics. | Fuzzy rules depend on the knowledge of subject expert and diagnosis accuracy depends on the available rules. Large amounts of rules and training data sets are required. |
| Bayesian networks | Abrupt faults [6,9,11–13] | Simultaneous multiple faults diagnosis. Graphic model easy to visualize and understand physical relationships. Information from data can be fused with expert knowledge. Coping with measurement uncertainty. Coping with missing information. Confidence levels (probability) are given for diagnostics results. | As the numbers of nodes and edges increase, the model complexity and computational requirements increase. High expert knowledge required for setting up the model. Knowledge of prior probability required, which can be difficult to assess. |
| Dynamic Bayesian networks | Abrupt faults [18] | All advantages of BNs. The prior probability is a dynamic function and can vary over time, making the problem more realistic. Interaction between multiple faults can be taken into account. | Same disadvantages as BNs, but easier estimation of prior probability. |

## 2. Methods

### 2.1. Bayesian Network

A Bayesian network is a graphical probabilistic model which represents the probabilistic relationship between events. It is an acausal model because it does not directly differentiate between causes and effects. One of the main components of the network is its graphical representation, in which events are depicted with nodes and relationships between events with arrows that connect the nodes to each other. Nodes from which arrows generate are called parent nodes (representing causes), and the nodes where the arrows end are child nodes (representing effects).

Another important component is the conditional probability table (CPT) for each node, which is a quantitative representation of the relationships between events. This includes each probability *P(X | Y)* that an event *X* occurs given that *Y* occurs. The posterior probability of the event *Y* given the observation *X* can then be calculated according to the Bayes theorem, as expressed in Equation (1).

$$P(Y|X) = \frac{P(X|Y)}{P(X)}P(Y) \tag{1}$$

Here, *P(Y)* represents the prior probability of *Y* and has to be known; *P(X)* is the marginal probability of the event *X*. In a fault classification problem, the event *X* can be, for example, a sensor measurement being too high, and the Bayes theorem relates this observation to the probability of a component being faulty. For fault identification, the event *Y* that we need to detect can be a specific fault severity. Both graphical structure and CPTs can be either constructed by experience or learned from data. Hybrid methods are also possible, where the human experience intervenes for incomplete or incorrect data. In this work, the BN structure was always built manually while the CPTs were fitted from data generated by a performance model by using the Expectation Maximization algorithm [28].

An advantage of BNs for diagnostics is that heterogeneous information can be fused together, such as gas path measurements, vibration measurements, and soft sensors. Another benefit is that the results are directly associated to a confidence level, which can be very useful for the operator to make decisions. There are, however, two limitations in the use of BNs for diagnostics of gas turbine systems, where several components usually age at the same time and other faults or abnormalities can occur on top of that. The first one is the size of the CPTs, which grows exponentially with a large number of nodes and states, making the problem intractable if we want to identify the correct fault and magnitude among several possible conditions. This limitation has been addressed by proposing hierarchical models [13,14] and will be tackled in Part 2 of this paper [27]. The second shortcoming concerns the choice of prior probability distribution, especially when sufficient data are not available from the system of interest. This issue will be analyzed in more detail in the next Section.

### 2.2. Prior Probability Distribution

Let *Y* be a random variable with two states, $y_1$ (healthy) and $y_2$ (faulty). We will analyze the effect of the prior probability on the detection of a faulty state for this binary example. The analysis can be extended to variables with multiple states, e.g., corresponding to different fault magnitudes.

For the Bayes theorem, the posterior probability of each state can be expressed according to Equations (2) and (3).

$$P(Y = y_1|X) = \frac{P(X|Y = y_1)P(Y = y_1)}{P(X)} \tag{2}$$

$$P(Y = y_2|X) = \frac{P(X|Y = y_2)P(Y = y_2)}{P(X)} \tag{3}$$

where $X$ is the observation (measurement residuals) and $P(X)$ the cumulative probability of the measurement residuals. We determine that the system is faulty if the condition in Equation (4) is true:

$$P(Y = y_2|X) > P(Y = y_1|X) \tag{4}$$

This condition holds true for bounded values of the prior probability $P(Y = y_2)$. Given that in a binary case, $P(Y = y_1) = 1 - P(Y = y_2)$, we can rewrite Equation (4) as Equations (5) and (6).

$$P(X|Y = y_2)P(Y = y_2) > P(X|Y = y_1)(1 - P(Y = y_2)) \tag{5}$$

$$P(Y = y_2) > \frac{Pr}{1 + Pr} \tag{6}$$

where $Pr$ is the probability ratio defined as $\frac{P(X|Y=y_1)}{P(X|Y=y_2)}$. This ratio is linked to the measurements' sensitivity to the fault. The numerator represents the probability to observe certain residuals in the measurements given a healthy system, which essentially depends on the measurement uncertainty. The denominator is instead the probability to observe the same residuals as a consequence of a fault. The $Pr$ is closer to 1 as the measurement errors increase and as the fault sensitivity decreases.

This means that, based on the probability ratio for a given observation $X$, the prior probability for $Y$ to be in state $y_2$ that results in a sufficient posterior probability is bounded by Equation (6), as illustrated in Figure 1.

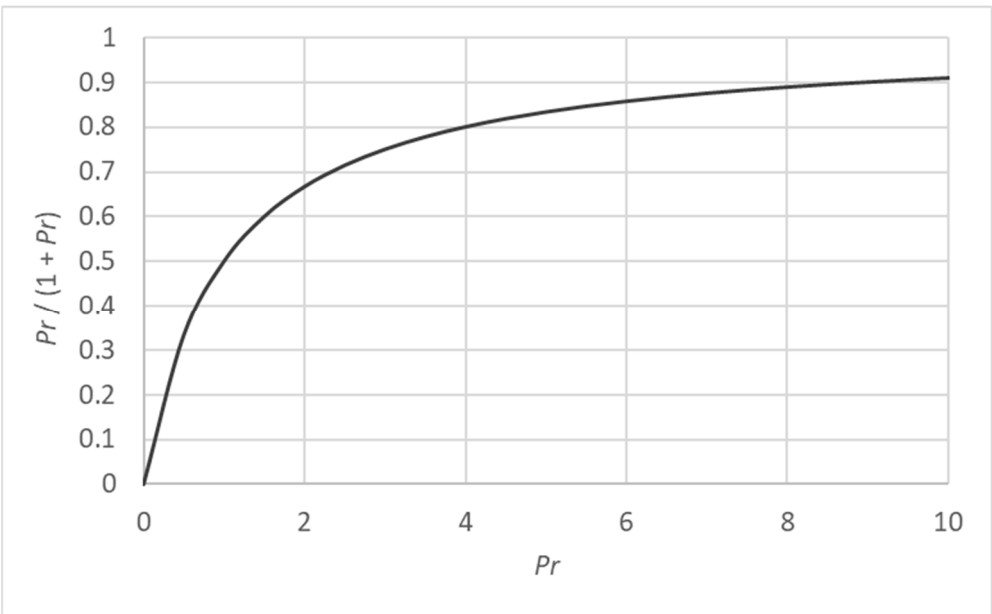

**Figure 1.** Threshold curve for the prior probability of a faulty state $y_2$. The area above the curve contains the values of $P(Y = y_2)$ that result in fault detection.

Usually, the prior probability of a faulty state is very low to reduce false alarms and also because it is often estimated from historical data, where the number of fault cases is limited. This means that a faulty state will be detected only for low values of $Pr$. As the uncertainty in the observation increases and the probability distributions of $X$ overlap, the prior belief of $y_2$ should be higher (e.g., >0.5 for $Pr = 1$).

Let us illustrate this with only one measurement, i.e., $X$ is a random variable representing the measured variable residual (i.e., distance from the nominal value). The distribution of $X$ for a healthy system has the mean in zero and a certain standard deviation $\sigma$ due to measurement noise and other process uncertainties. We assume a Gaussian distribution, without loss of generality. In a faulty state, the mean of $X$ is shifted by a certain amount

Δ, depending on the sensitivity of the measured variable to the fault [11]. We can express Δ as a function of σ, e.g., 2σ, 3σ, 4σ, etc., depending on how sensitive the measurement is to the fault.

The goal in this analysis is to assess the probability ratio $Pr$ as a function of $\Delta = k\sigma$. For a continuous Gaussian function, the probability for a certain value of $x$ is expressed in Equation (7).

$$p = \frac{1}{\sigma\sqrt{2\pi}} \exp\left(-\frac{1}{2}\left(\frac{x-\mu}{\sigma}\right)^2\right) \tag{7}$$

Therefore, with the previous assumptions, we can rewrite the probability ratio as in Equation (8).

$$Pr = \frac{\exp\left(-\frac{1}{2}\left(\frac{x}{\sigma}\right)^2\right)}{\exp\left(-\frac{1}{2}\left(\frac{x-k\sigma}{\sigma}\right)^2\right)} \tag{8}$$

Normalizing by $x/\sigma$, we obtain the following graph for various values of $k$, shown in Figure 2.

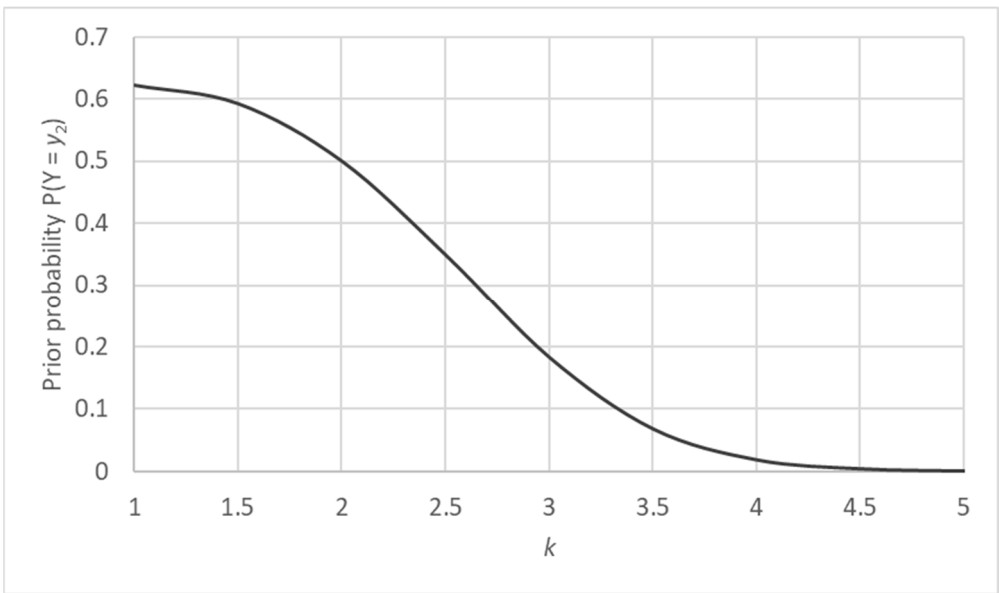

**Figure 2.** Threshold curve for the prior probability of a faulty state $y_2$ as a function of the measurement sensitivity $k$.

On the $y$-axis, the graph displays the lower limit for the prior probability of $y_2$ that is necessary to detect the faulty state given an observed residual $x = k\sigma - \sigma$. It is evident that for low values of $k$ (i.e., smaller deviations compared to the measurement uncertainty), the prior probability has to be quite high to detect a faulty condition from an observation within one standard deviation from the mean. The curve is of course obtained from continuous Gaussian distributions; hence, it is to be used only as a qualitative indication and cannot be directly transposed to other data distributions.

A trade-off is present, however, due to the concurrent aim to minimize false alarms, which requires us to set a lower probability for rare faulty events. Therefore, this analysis highlights the need for a dynamic prior probability distribution that is updated over time.

*2.3. Dynamic Bayesian Network*

In a dynamic Bayesian network (DBN), the prior probability $P(Y)$ is not constant, but it depends on some hyperparameter $\varphi$ as generalized in Equation (9) for a continuous distribution [25].

$$P(Y) = \int_\Phi P(Y|\varphi)P(\varphi)d\varphi \tag{9}$$

The prior $P(Y|\varphi)$ can be, for example, a function of time, previous conditions, or maintenance actions. With first-order Markov assumptions, the state variables at time $t$ depend only on the variables at time $t-1$ (forgetting assumption). This is illustrated by Equation (10).

$$P(X_t|X_{0:t-1}) = P(X_t|X_{t-1}) \tag{10}$$

For example, if the compressor's previous condition is known to be healthy, the prior probability of the event $Y$ = 'Healthy' should be very high; on the contrary, if the previous detected condition is fouling, the event $Y$ = 'Healthy' should have a probability close to zero, and the event $Y$ = 'Fouling' should be much more likely since the conditions can only get worse over time.

In this work, two BNs were developed to identify gradual deterioration in the compressor and in the turbine, respectively, with the possibility of simulating a static BN or a dynamic one. The general structure is shown in Figure 3. This structure is assumed to be the same at each time interval. The child nodes are the measurement residuals and are composed of eight discrete states each, from very very low (VVL—negative residuals) to very very high (VVH—positive residuals). These states represent non-overlapping intervals of measurement deviations from the reference value and were selected by dividing the span between minimum and maximum deviations into 8 intervals so that the state corresponding to no deviation lies in an interval approximately equal to $\pm 3\sigma$ (sensor noise).

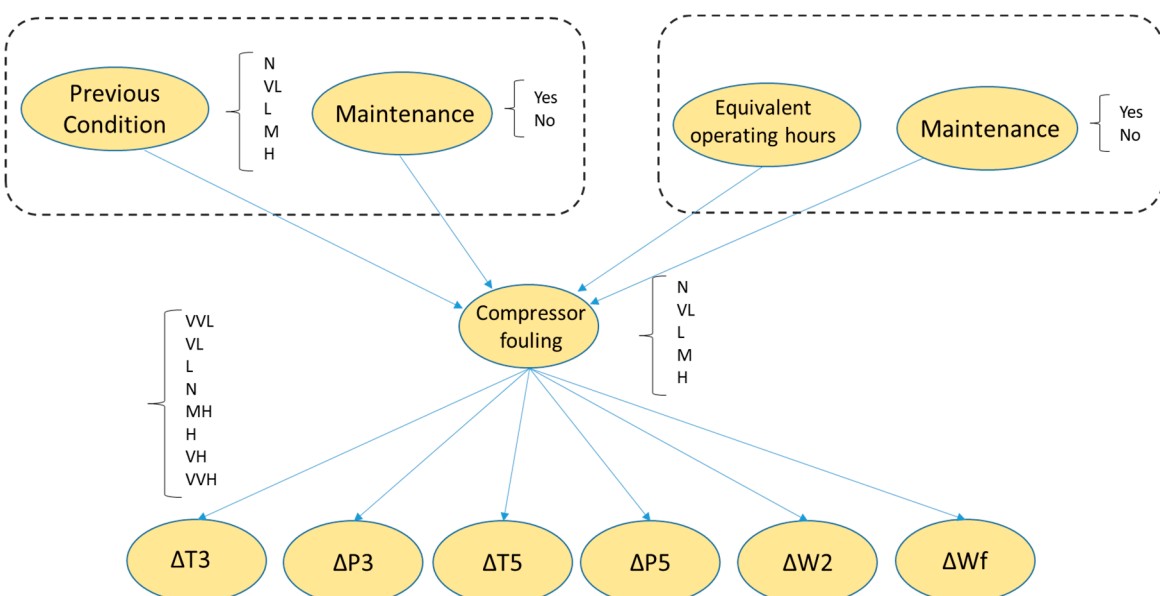

**Figure 3.** The structure of the BN (the node for compressor fouling is replaced with turbine erosion for the second network). The nodes in the dashed areas are added to simulate a DBN where $P(Y)$ depends either on the previous condition or on time. Maintenance = Yes resets the $P(Y)$ distribution to healthy conditions.

Unlike the example of Section 2.2, the observation $X$ in this BN includes the probability distributions of 6 variables. The probability $P(X|Y)$ can be expressed as the product of the individual probabilities for each variable (i.e., measurement residual); these are represented in a discrete network by the elements of the CPTs.

The parent nodes represent compressor condition and turbine condition in the two BNs. These were considered as a function of some deviation factors in component efficiency and flow capacity. The deviation factors were calculated as per Equations (11) and (12).

$$\Delta \eta = \eta_{fault} - \eta_{healthy} \tag{11}$$

$$\Delta \overline{W} = \frac{\overline{W}_{fault}}{\overline{W}_{healthy}} \tag{12}$$

The parent node (compressor or turbine condition) included 5 states:

- Normal conditions (N)—fault severity equal to zero.
- Very low degradation (VL)—fault severity lower than 1%, which together with N represents healthy conditions.
- Low degradation (L)—fault severity between 1% and 2%.
- Medium degradation (M)—fault severity between 2% and 3%.
- High degradation (H)—fault severity higher than 3%.

The fault severity $S$ was a function of efficiency and flow capacity deviation calculated according to Simon [29] as in Equation (13). This work focused on small to medium fault severity magnitude because they are usually more difficult to detect. Fault severities higher than 3% could also be included by increasing the number of states in the parent nodes.

$$S = -\Delta \eta \cdot \sqrt{1 + \left( \frac{\Delta \overline{w} - 1}{\Delta \eta} \right)^2} \tag{13}$$

Each network was trained with data generated by a performance model described in Section 2.4, including different levels of deviation in efficiency and flow capacity, different ratios $\frac{\Delta \overline{w} - 1}{\Delta \eta}$ between 1 and 2, and measurements noise. The training phase was used to estimate the CPTs for each node by using the Expectation Maximization algorithm [28]. For the DBN, a time-dependent function was assigned to the prior probability $P(Y)$ according to two different models: first, a Poisson distribution as a linear function of time was tested, and secondly, a first-order Markov model as a function of the previous condition was selected. Both cases were compared to a static BN with fixed $P(Y)$.

It is important to note the effect of noise and engine-to-engine variations on the state distribution in the CPTs. Since the BN should be as general as possible and applicable to the whole fleet, engine-to-engine variations have to be taken into consideration. Initial variations in efficiency and flow capacity due to production deviations were considered as $\pm 0.5\%$ and $\pm 1\%$, respectively [11]. Further, degradation phenomena such as fouling and erosion may manifest with different ratios between flow capacity and efficiency in different machines. Figure 4 exemplifies this for the case of compressor fouling. Due to different fouling ratios, production scatter, and measurement noise, there is a large overlap of the states VL and L and the states L and M.

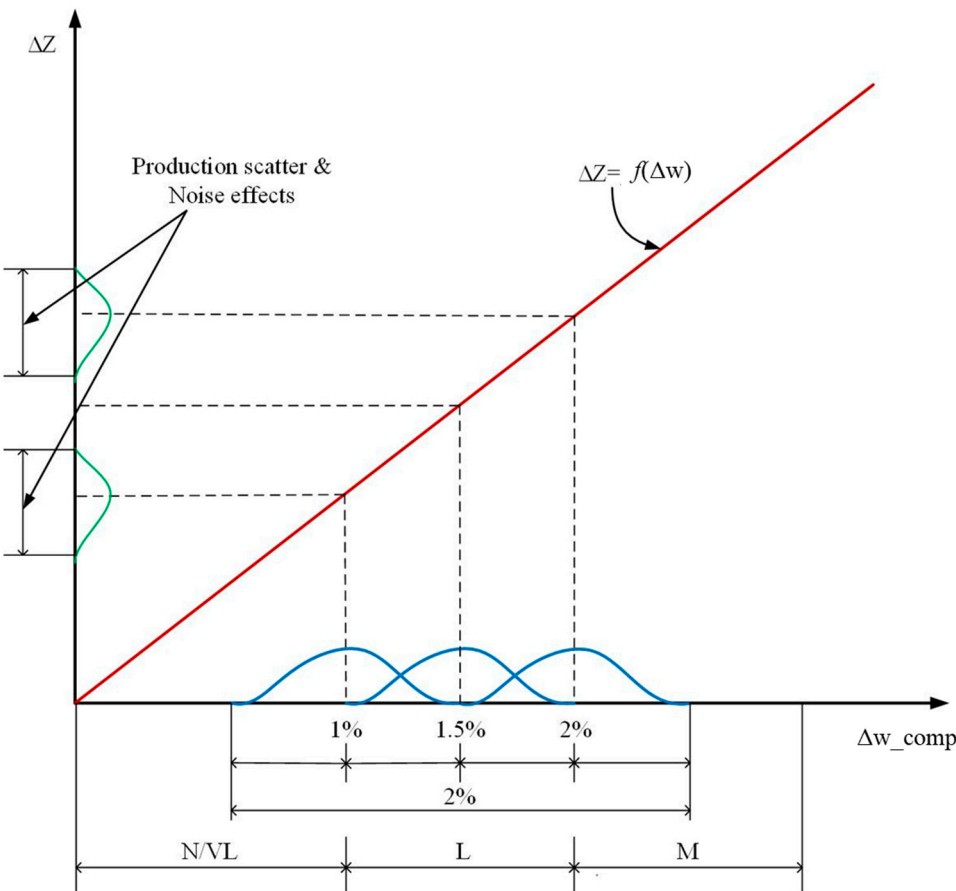

**Figure 4.** Example of states overlap for compressor fouling.

## 2.4. Gas Turbine Model

A performance model was used to generate synthetic data to train and test the BNs. The performance model of the gas turbine system has been extensively described in previous work [11,30–32]. Compared to [11,30], and similarly to [32], the model was modified to simulate a single-shaft gas turbine connected to a generator. The gas turbine under consideration is a single-shaft 50 MW class turbine such as the Siemens Energy SGT800. The validity of the model was proven against reference data from the engine of interest in [32].

The measurements used in this work are: the compressor outlet temperature (T3) and pressure (P3); the compressor inlet flow rate (W2), which is measured through a bellmouth intake; turbine exhaust temperature (T5) and pressure (P5); and fuel flow (Wf). The schematic of the gas turbine system and the sensors' locations is presented in Figure 5.

The model inputs are the requested power, ambient conditions (temperature, pressure, and relative humidity), and IGV position. In addition, performance deviation factors as defined in Equations (11) and (12) are given as input data and used to adjust the performance maps. The outputs are the measurements residuals, calculated according to Equation (14).

$$ r = \frac{z - z_{ref}}{z_{ref}} \tag{14} $$

In this Part 1, compressor fouling and turbine erosion are simulated separately, i.e., simultaneous degradation is not considered. In Part 2, simultaneous degradation of the rotating components and concurrent presence of abrupt faults will be addressed.

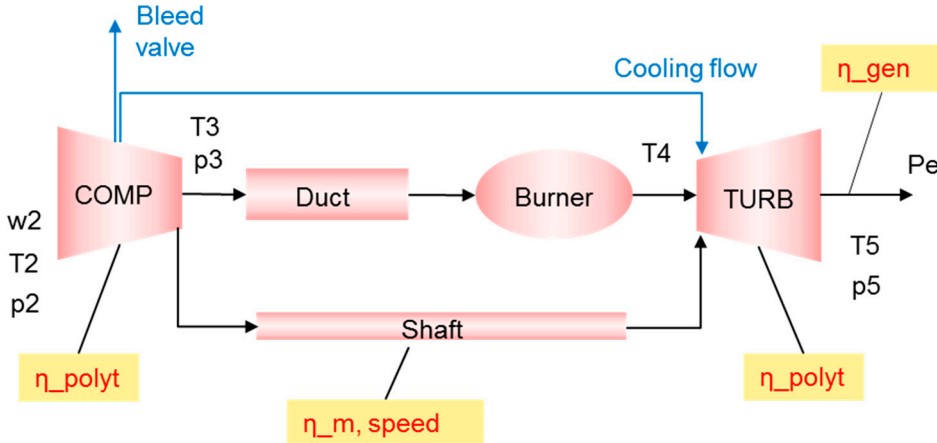

**Figure 5.** Schematic diagram of the gas turbine model (η_polyt: polytropic efficiency, η_m: shaft mechanical efficiency, η_gen: generator efficiency).

## 3. Results and Discussion

Gradual compressor fouling and turbine erosion were simulated with the gas turbine model. The sensitivity of each measurement residual to a fault severity L (i.e., between 1% and 2%) is presented in Tables 2 and 3. As stated in Section 2.3, fault severity below 1% (corresponding to, for example, 0.5% efficiency deviation and 1% flow capacity deviation) is generally labelled as a healthy condition. This is because such low severity cannot be distinguished from manufacturing tolerance or measurement noise. Therefore, healthy data present a standard deviation due not only to sensor noise but also to small performance degradation not yet considered a fault. Note that the quantity Δ was calculated as the difference between the mean values of the healthy and faulty data distributions.

**Table 2.** Compressor fouling data distribution (Fault severity L).

| Sensor | Δ | $\sigma$ | $k = \Delta/\sigma$ |
|---|---|---|---|
| T3 | 0.0022 | 0.0008 | 2.75 |
| P3 | 0.0054 | 0.0017 | 3.17 |
| T5 | 0.0089 | 0.002 | 4.4 |
| P5 | 0.00004 | 0.00001 | 4.0 |
| W2 | 0.0082 | 0.002 | 4.1 |

**Table 3.** Turbine erosion data distribution (Fault severity L).

| Sensor | Δ | $\sigma$ | $k = \Delta/\sigma$ |
|---|---|---|---|
| T3 | 0.0014 | 0.00026 | 5.38 |
| P3 | 0.0055 | 0.00098 | 5.6 |
| T5 | 0.013 | 0.0021 | 6.26 |
| P5 | 0.000067 | 0.000011 | 6.0 |
| W2 | 0 | 0.0018 | 0 |

As can be observed from the above Tables, the simulated compressor fouling appeared with smaller relative deviations *k* than turbine erosion. This is particularly true for T3 and P3, which are the measurements more affected by compressor degradation. A high standard deviation is observed for these two measurements in healthy conditions.

From Table 2, large deviations in the variables T3, P3, and W2 are evident, as it is known that compressor fouling leads to reduced pressure ratio, efficiency, and flow capacity. Because of the high sensitivity of these variables, even small fault severity, which is not detected by a diagnostic system, contributes to increased scatter in the healthy data.

Turbine erosion (Table 3) appears as a larger deviation in T5 and P5 due to decreased turbine efficiency. Since the turbine is choked, we do not see any deviation in W2 despite the increased flow capacity. Overall, the recorded standard deviations for the measured variables are lower in Table 3, which leads to higher values of *k*.

These represent two perfect cases to assess the effect of prior probability distribution, as for the compressor, we have a case with $k < 4$ overall and ~3 for the most sensitive residuals; and for the turbine, $k > 5$ overall, including the most sensitive residuals. The three different approaches for BN were tested on two simulated linear degradation patterns for compressor and turbine, respectively. The degradation severity was assumed to increase linearly from 0% to 3.6% (i.e., from N to H) in 8000 h. Note that this choice was arbitrary, only for test purposes, and does not represent actual degradation of the SGT800 or any real engine. In the following Sections, three cases are presented and compared: a static BN where the prior probability distribution in the parent nodes is constant over time, a DBN where the prior probability distribution evolves over time following a Poisson distribution, and a DBN where the parent nodes are constructed as temporal nodes, i.e., conditionally dependent on the condition in the previous time step.

### 3.1. Constant Prior Distribution

In case of constant prior probability distribution, the maximum probability should be associated with the healthy state to avoid false alarms. Degraded states may have the same probability if one assumes that, once degradation commences, the component will experience linearly growing degradation until a corrective action occurs. To the best of the authors' knowledge, there are no guidelines or best practices in the open literature on suitable distributions. One has to rely on historical data or the experience of the operators. The probability distribution selected for this work is illustrated in Table 4. N and VL conditions together account for 97% of probability, which is for healthy conditions. Since the prior probability of a faulty state is 1%, we can expect that cases with low *k* values (large noise or uncertainty, or small measurement sensitivity to the fault) will be misclassified. The results in Table 4, for compressor fault, confirm this hypothesis.

**Table 4.** Prior probability $P(Y)$ for compressor fouling and turbine erosion.

| N | VL | L | M | H |
|---|----|---|---|---|
| 90% | 7% | 1% | 1% | 1% |

For compressor fouling, we see in Table 5 a 20% instance of false alarms, which may prevent any type of forecasting of degradation evolution and predictive maintenance. A similar observation can be made for the 11% wrongly predicted cases of medium severity. If the threshold for compressor washing is, for instance, set as level H, the static BN would cause a premature maintenance action. The overall accuracy for the BN predictions was about 84%.

**Table 5.** Confusion matrix for a static BN for compressor fouling, $P(Y)$ = const.

| | Predicted | N/VL | L | M | H |
|---|---|---|---|---|---|
| | N/VL | 80% | 20% | 0% | 0% |
| Real | L | 0% | 89.5% | 10.5% | 0% |
| | M | 0% | 0% | 89% | 11% |
| | H | 0% | 0% | 1% | 99% |

Table 6 shows that the prediction accuracy for the turbine erosion is lower when it comes to distinguishing healthy conditions from L severity level, with 12% false negatives with low degradation. On the contrary, only 0.5% false alarms occurred. The overall accuracy of the BN was 90%, as expected from the data distributions presented in Table 3 (i.e., states were more clearly distinct than for the compressor).

**Table 6.** Confusion matrix for a static BN for turbine erosion, $P(Y)$ = const.

| | Predicted | N/VL | L | M | H |
|---|---|---|---|---|---|
| **Real** | N/VL | 99.5% | 0.5% | 0% | 0% |
| | L | 12% | 73% | 14% | 0% |
| | M | 0% | 0% | 90% | 10% |
| | H | 0% | 0% | 0% | 100% |

### 3.2. Time-Dependent Prior Distribution

One way to improve the fault probability estimation is to consider that "older" machines are more likely to be in a degraded state than new machines. The prior probability for each state can be assigned as a function of the equivalent operating hours according to the history of the machine and the experience of the operators. In [26], a discrete Poisson distribution was used to assess the prior probability of failure rate. In this case, we are not interested in the failure rate as a function of the number of events, but we can nonetheless use the same function where the event number is replaced by the health state $Y_i$ (from 0 for N to 4 for H). The discrete distribution is described by Equation (15), where the coefficient $\lambda$ is varied linearly with time, i.e., $\lambda = dt + \lambda_0$, where $d$ represents the degradation rate and $\lambda_0$ the initial value for $\lambda$.

$$p(x) = \frac{\lambda^{Y_i}}{Y_i!} e^{-\lambda} \tag{15}$$

It was assumed that an estimated degradation rate is known from historical data, in this case 3.6% over 8000 h, and the time interval was divided equally among the fault severity values, i.e., degradation was considered linear. Since $\lambda$ is the mean value, it was varied accordingly with the expected average conditions of the machine at a given time. The resulting probability distribution for some discrete time intervals is illustrated in Figure 6, and values are reported in Table 7. It is possible to note that the highest probability is shifted toward a higher severity level over time, and the standard deviation is also increasing. The first effect should promote a better fault identification for low $k$ values, but the latter may, on the contrary, limit the accuracy.

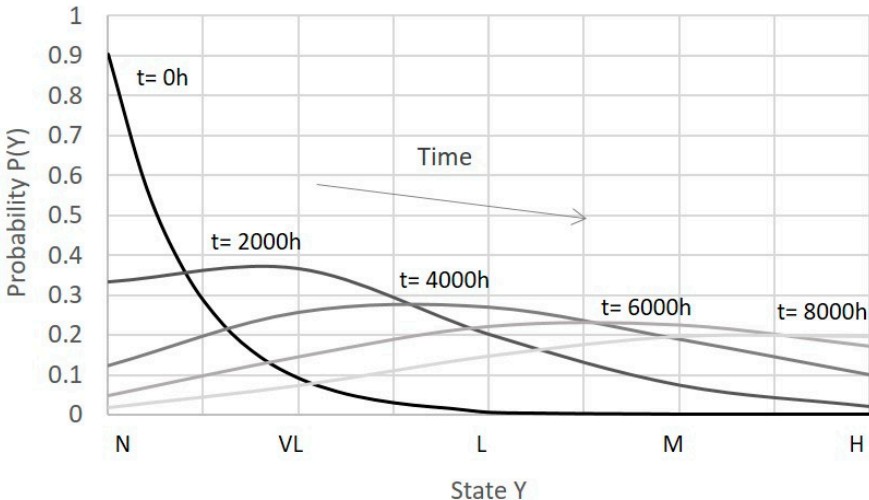

**Figure 6.** Poisson probability distributions $P(Y)$ over operating time.

**Table 7.** Prior probability $P(Y)$ as a function of time (or equivalent operating hours).

|      | 0 h   | 2000 h | 4000 h | 6000 h | 8000 h |
|------|-------|--------|--------|--------|--------|
| N    | 90.5% | 33%    | 12%    | 5%     | 1.8%   |
| VL   | 9%    | 37%    | 25%    | 15%    | 7%     |
| L    | 0.5%  | 20%    | 27%    | 22%    | 15%    |
| M    | 0%    | 7%     | 18%    | 23%    | 19%    |
| H    | 0%    | 2%     | 10%    | 16%    | 20%    |

The classification accuracy did not increase substantially compared to the static BN, as shown in Tables 8 and 9. Because of the characteristics of Poisson distribution, the standard deviation in the prior distribution increases over time. Therefore, even if the maximum probability shifted from N to H, the results were very similar to the fixed prior distribution case. A reduction in false H severity alarms for CF was observed, which can be beneficial for maintenance scheduling. However, a higher false alarm rate for L severity decreased the overall network accuracy to 81%. A general improvement was observed for TE diagnosis, with some misdiagnoses for medium degradation that did not occur with the static BN. A different distribution could be selected and results could be improved on a case-to-case basis, but the proposed solution is meant to be generic and easily applied to any machine. It is important to remember, however, that these results are obtained assuming to know the evolution of degradation phenomena from historical data. This is not always possible, and therefore, this approach cannot be a recommended solution for general cases.

**Table 8.** Confusion matrix for a DBN for compressor fouling, $P(Y) = f(t)$.

|      | Predicted | N/VL | L   | M     | H    |
|------|-----------|------|-----|-------|------|
| Real | N/VL      | 74%  | 26% | 0%    | 0%   |
|      | L         | 0%   | 90% | 10%   | 0%   |
|      | M         | 0%   | 0%  | 93.8% | 6.2% |
|      | H         | 0%   | 0%  | 0%    | 100% |

**Table 9.** Confusion matrix for a DBN for turbine erosion, $P(Y) = f(t)$.

|      | Predicted | N/VL | L   | M   | H    |
|------|-----------|------|-----|-----|------|
| Real | N/VL      | 96%  | 4%  | 0%  | 0%   |
|      | L         | 0%   | 88% | 12% | 0%   |
|      | M         | 0%   | 0%  | 90% | 10%  |
|      | H         | 0%   | 0%  | 0%  | 100% |

### 3.3. Condition-Based Prior Distribution

For the third case, the parent nodes were built as temporal nodes, and a CPT for the nodes "compressor condition" and "turbine condition" as a function of the previous state (as illustrated in Figure 3) was estimated during the training phase. The resulting probability distributions for the temporal nodes are illustrated in Figure 7, and the values are reported in Table 10.

As shown in Table 10, the maximum probability is assigned to be in the same condition as the previous one. This is because phenomena such as compressor fouling and turbine erosion evolve gradually, and if we evaluate the conditions on an hourly or even daily basis, we do not expect to see a significant change. For the same reason, we do not expect to jump from a condition of very low degradation to high degradation in one step; therefore, the probability of state H given the previous state VL or L is zero. In fact, any sudden change in performance has to be attributed to an abrupt fault, which should not be detected by this network. Additionally, in reality, the degradation severity can only increase and never decrease. However, a 1% probability to step back to a previous condition was allowed to avoid excessive false alarms due to measurement noise. This is particularly true

for low degradation scenarios where the noise can occasionally appear as a deviation in performance; if this happens for a single data point and it does not show as a persistent deviation, it has to be considered as an outlier. Looking at the parameters in Table 10, we can expect to improve fault identification for the cases with low *k* values since the maximum probability is shifted to higher fault severity over time and the standard deviation in the probability distribution remains the same.

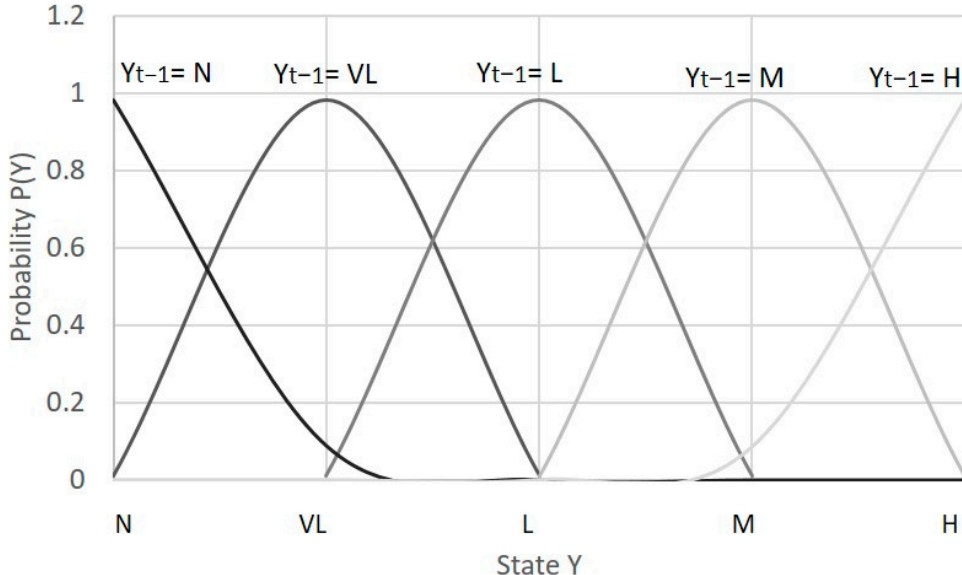

**Figure 7.** Prior probability distributions *P(Y)* for different previous conditions.

**Table 10.** Prior probability *P(Y)* for CF and TE as a function of previous conditions.

| Previous Condition | N | VL | L | M | H |
|---|---|---|---|---|---|
| N | 99% | 1% | 0% | 0% | 0% |
| VL | 1% | 98% | 1% | 0% | 0% |
| L | 0% | 1% | 98% | 1% | 0% |
| M | 0% | 0% | 1% | 98% | 1% |
| H | 0% | 0% | 0% | 1% | 99% |

The results of the two DBNs are presented in Tables 11 and 12 and indicate that a considerable improvement can be achieved for CF diagnosis. Some misclassifications are still present, e.g., due to noise in the data, but the false alarms are entirely avoided. Quite surprisingly, the accuracy of the DBN for healthy turbine conditions was slightly lower than with a static BN, with 5% of data points incorrectly classified as L. Nonetheless, the overall accuracy was 5 percentage points higher than for the static BN. These results are very promising when considering the significant scatter in the data and states overlap due to engine-to-engine variations and measurement noise. The improvement reached with dynamic BNs can be summarized as in Table 13.

**Table 11.** Confusion matrix for a DBN for compressor fouling, $P(Y) = f(Y_{t-1})$.

| | Predicted | N/VL | L | M | H |
|---|---|---|---|---|---|
| | N/VL | 100% | 0% | 0% | 0% |
| Real | L | 2.7% | 88% | 9.3% | 0% |
| | M | 0% | 0% | 100% | 0% |
| | H | 0% | 0% | 0% | 100% |

**Table 12.** Confusion matrix for a DBN for turbine erosion, $P(Y) = f(Y_{t-1})$.

| | Predicted | N/VL | L | M | H |
|---|---|---|---|---|---|
| Real | N/VL | 95% | 5% | 0% | 0% |
| | L | 0% | 94% | 6% | 0% |
| | M | 0% | 0% | 100% | 0% |
| | H | 0% | 0% | 0% | 100% |

**Table 13.** Overall diagnostics accuracy for the three BN approaches.

| | $P(Y)$ = constant | $P(Y) = f(t)$ | $P(Y) = f(Y_{t-1})$ |
|---|---|---|---|
| Compressor | 84% | 81% | 92% |
| Turbine | 90% | 93% | 95% |

These results prove that, for faults characterized by low measurement sensitivity compared to the scatter present in non-faulty data, a dynamic Bayesian network that tracks previous conditions can be a successful approach to minimize misdiagnoses. On the contrary, with limited standard deviation in non-faulty data or high sensitivity to faults, the improvement given by a DBN compared to a static BN is not significant, as the information carried by the conditional probability term $P(X | Y)$ is dominant over the prior probability $P(Y)$. The DBN based on equivalent operating hours showed a slight improvement compared to a static BN for the TE, but a drawback is that it necessitates knowledge on the expected degradation rate of the component. Since this information is seldom available, the first-order Markov model for the prior probability as a function of the previous condition is recommended as a suitable solution, which also showed the best results among the tested networks.

## 4. Conclusions

Dynamic Bayesian networks offer a promising solution for diagnosis of gas turbine degradation in the presence of uncertain data. Compared to previous demonstrations, in this work, the Bayesian network (BN) was trained with simulated fleet data containing engine-to-engine variations in healthy and degraded operations. A dynamic update of the component conditions' prior probability was assessed, and three approaches were compared: (1) static BN with fixed prior distribution, (2) dynamic BN (DBN) with a time-variant Poisson prior distribution, and (3) a DBN with a hyperprior based on previous component conditions. It was shown that the condition-based DBN resulted in the best accuracy, and the benefit was more significant for data with higher overlap between states (i.e., for compressor fouling). The improvement in classification accuracy with the proposed DBN was 8 percentage points for compressor fouling and 5 points for turbine erosion compared with the static BN. In this Part 1 of the work, only one degrading component at a time was considered. Part 2 will demonstrate the successful application of the proposed method on a real engine by integrating the BN models and testing on field data to discern simultaneous gradual degradation in multiple components and abrupt faults [27].

**Author Contributions:** Conceptualization, V.Z., A.D.F., and K.K.; methodology, V.Z.; formal analysis, V.Z. and A.D.F.; writing—original draft preparation, V.Z.; writing—review and editing, A.D.F. and K.K.; project administration, K.K. All authors have read and agreed to the published version of the manuscript.

**Funding:** This research was funded by the Swedish Knowledge Foundation (KKS) under the project PROGNOSIS, grant number 20190994.

**Institutional Review Board Statement:** Not applicable.

**Informed Consent Statement:** Not applicable.

**Data Availability Statement:** Not applicable.

**Acknowledgments:** The authors would like to acknowledge the other researchers in the project as Anna Sjunnesson and Andreas Hansson from Siemens for their support, as well as Mikael Stenfelt and Xin Zhao for their contribution to the performance model development.

**Conflicts of Interest:** The authors declare no conflict of interest. The funders had no role in the design of the study; in the collection, analyses, or interpretation of data; in the writing of the manuscript; or in the decision to publish the results.

## Nomenclature

*Acronyms*

| | |
|---|---|
| BN | Bayesian Network |
| CF | Compressor Fouling |
| CPT | Conditional Probability Table |
| DBN | Dynamic Bayesian Network |
| H | High |
| IGV | Inlet Guide Vane |
| L | Low |
| M | Medium |
| N | Normal |
| TE | Turbine Erosion |
| VH | Very High |
| VL | Very Low |

*Symbols and Greek letters*

| | |
|---|---|
| $P$ | Probability distribution |
| $Pr$ | Conditional probability ratio |
| $r$ | Residual |
| $S$ | Fault severity |
| $\overline{w}$ | Flow capacity |
| $z$ | Measurement |
| $\Delta$ | Deviation from healthy conditions |
| $\eta$ | Efficiency |
| $\lambda$ | Poisson coefficient |
| $\sigma$ | Standard deviation |
| $\varphi$ | Hyperparameter |

*Subscripts*

| | |
|---|---|
| ref | Reference conditions |
| t | Time |

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
