# Peer review of "Assessment of Dynamic Bayesian Models for Gas Turbine Diagnostics, Part 1: Prior Probability Analysis"

_machines, doi:10.3390/machines9110298_

Round 1

Reviewer 1 Report

  1. In the results and discussion section, the authors are requested to describe the conditional parameters and their implications more clearly, used in the Tables.
  2. The three approaches used in the paper must be depicted with their differences
  3. In the results section and tables, it is quite difficult to identify the approaches used. Can u make it more obvious?
  4.  A graphical presentation of the comparison is recommended.  
  5. Can u also briefly describe how BN compares with other methods 
  6. Where is the field data coming from?

Author Response

Dear Reviewer, thank you for the valuable comments, we believe helped us improve our manuscript. Please find a detailed response in the attached file.

Reviewer 2 Report

The manuscript proposes two Bayesian networks were for compressor fouling and turbine erosion diagnostics. Authors have compared different prior probability distributions were compared to determine the benefits of a dynamic, first-order hierarchical Markov model over a static prior probability and one dependent only on time. 

The main idea of the paper is interesting and fits the aims and scope of the journal. The methodology is presented very well and the results are reproducible for readers in the field. I believe that the paper could be considered for publication in the Machines after addressing the following minor rooms by the authors: 

1- The main idea of the literature review should be a critical gap analysis to justify the necessity of doing this piece of research. The limitations, assumptions, and relative merits of the proposed approach to other publicly available proposals should be highlighted more clearly. 

2- There are some minor typos and grammatical errors in the manuscript. So, the authors are recommended to go through their work once again from the English language point of view. 

Author Response

Dear Reviewer, thank you for taking the time to read through our manuscript and provide your insights. We have improved the literature review by adding several references and discussion around the gap. We have also carefully checked the language and corrected some minor spelling mistakes. We hope our changes are satisfactory.

Reviewer 3 Report

The manuscript assets the use of a Dynamic Bayesian Network  for diagnostics of a power generation gas turbine, considering highly-scattered fleet data to train the network.

A dynamic update of the component conditions prior probability was assessed, and three approaches were compared: 1) static Bayesian Network   with fixed prior distribution, 2) Dynamic Bayesian Network   with a time-variant Poisson prior distribution, and 3) Dynamic Bayesian Network  with a hyperprior based on previous component condition.

The confusion matrices (predicted vs. real) show that the condition-based Dynamic Bayesian Network   resulted in the best accuracy, and the benefit is more significant for data with higher overlap between states.

The manuscript is well structured and documented, and it is interesting for the readers.

The pursued objectives are well defined and the conclusions are supported by results.

Author Response

Dear Reviewer, thank you for taking the time to read through our manuscript and provide your insights.